

# DDX17 modulates the expression and alternative splicing of genes involved in apoptosis and proliferation in lung adenocarcinoma cells

Cheng He[1,2], Gan Zhang[3], Yanhong Lu[3], Jingyue Zhou[3] and Zixue Ren[3]

[1] Department of Thoracic Oncology, The First Affiliated Hospital of University of Science and Technology of China, Hefei, Anhui, China
[2] Department of Thoracic Oncology, Anhui Provincial Cancer Hospital, Hefei, Anhui, China
[3] Department of Thoracic Surgery, Anhui Provincial Cancer Hospital, Hefei, Anhui, China

## ABSTRACT

**Background**. The DEAD-box RNA-binding protein (RBP) DDX17 has been found to be involved in the tumorigenesis of many types of cancers. However, the role of DDX17 in lung adenocarcinoma (LUAD) remains unclear.

**Methods**. We silenced DDX17 expression in A549 LUAD cells by small interfering RNA (siRNA). Cell proliferation and apoptosis assays were performed to explore the functions of DDX17. Knockdown of DDX17 by siRNA significantly inhibited proliferation and induced apoptosis in A549 cells. We used high-throughput RNA sequencing (RNA-seq) to identify differentially expressed genes (DEGs) and alternative splicing (AS) events in DDX17 knockdown LUAD cells.

**Results**. DDX17 knockdown increased the expression levels of proapoptotic genes and decreased those of proproliferative genes. Moreover, the DDX17-regulated AS events in A549 cells revealed by computational analysis using ABLas software were strongly validated by quantitative reverse transcription–polymerase chain reaction (RT–qPCR) and were also validated by analysis of The Cancer Genome Atlas (TCGA)-LUAD dataset. These findings suggest that DDX17 may function as an oncogene by regulating both the expression and AS of proliferation- and apoptosis-associated genes in LUAD cells. Our findings may offer new insights into understanding the molecular mechanisms of LUAD and provide a new therapeutic direction for LUAD.

# INTRODUCTION

Lung cancer remains the leading cause of cancer-related death worldwide (*Sung et al., 2021*). Non-small-cell lung cancer (NSCLC) accounts for approximately 85% of all lung cancers, and lung adenocarcinoma (LUAD) is the most common type of NSCLC. Recently, progress has been made in understanding the molecular mechanisms of lung cancer, which is critical for lung cancer therapy. For example, the epidermal growth factor receptor (EGFR) signaling pathway plays an important role in cell growth, proliferation and survival (*Sibilia et al., 2007*). The Iressa Pan-Asia Study (IPASS) showed that EGFR tyrosine kinase

Corresponding author
Zixue Ren,
renzixue1975@outlook.com

inhibitors (EGFR-TKIs) significantly prolonged the survival of NSCLC patients harboring EGFR mutations, a finding that revolutionized the treatment of lung cancer (*Chan & Hughes, 2015*). Tumor immune escape is an important mechanism promoting tumor development (*Liu et al., 2020*). Programmed death ligand 1 (PD-L1), a molecule in the B7 family, has been implicated in tumor immune escape. Immunotherapies for lung cancer, such as PD-1/PD-L1 monoclonal antibodies, are currently a clinical research focus. Despite remarkable and promising progress in treatments for lung cancer, such as targeted therapy and immunotherapy, the survival rate of patients with lung cancer at 5 years after diagnosis is approximately 20% in most countries (*Sung et al., 2021*). Therefore, new strategies to improve clinical outcomes for NSCLC patients must be explored.

Increased attention has been given to the role of RNA-binding proteins (RBPs) in carcinogenesis (*Pereira, Billaud & Almeida, 2017*). RBPs can regulate every aspect of RNA biology, including transcription, splicing, modification, and translation, all of which influence gene and protein expression (*Gebauer et al., 2021*). Therefore, minor changes in RBPs may lead to significant alterations in gene expression, which can be a cause of human diseases, including cancer. Many studies have demonstrated that RBPs can act as either tumor suppressors or oncogenes in lung cancer (*Bitaraf et al., 2021*; *Shen et al., 2020*; *Sarkar & Ghosh, 2016*).

DEAD-box RNA helicase 17 (DDX17), a DEAD-box RNA helicase, is an RBP that participates in DNA repair, histone modification and microRNA (miRNA) regulation (*Kao et al., 2019*; *Wu, 2020*). DDX17 can recognize and directly bind specific regions in its target RNAs and remodel their structures (*Ngo, Partin & Nam, 2019*). DDX17 is involved in several cell signaling pathways that are crucial for survival and apoptosis by regulating gene expression and oncogenesis (*Sarkar & Ghosh, 2016*). Posttranslational modification of DDX17 may lead to transcriptional activation of cancer stemness-related genes (*Wu, 2020*). Aberrant DDX17 expression has been found in hepatocellular carcinoma (HCC), colon cancer, glioma and lung cancer (*Xue et al., 2019*; *Shin et al., 2007*; *Luo et al., 2020*; *Li et al., 2017a*).

In addition, alternative splicing (AS), a posttranscriptional modification process, plays a crucial role in the production of diverse mature messenger RNA (mRNA) transcripts (*Peng et al., 2021*). It is currently believed that the expression of approximately 95% of human genes is regulated by AS (*Le et al., 2015*). In addition, accumulated evidence indicates that abnormal AS events (ASEs) frequently occur during the progression and development of many cancers, including LUAD (*Deng et al., 2021*; *Bonomi et al., 2013*; *Coomer et al., 2019*). DDX17 can regulate AS of PXN-AS1 in HCC to promote tumor metastasis (*Zhou et al., 2021*). However, the regulatory role of DDX17 in gene expression and AS in LUAD is incompletely defined.

To study the potential function of DDX17 in regulating gene expression and AS, which might be related to LUAD, we obtained DDX17-regulated transcriptomes in human LUAD cells (A549 cells) by RNA sequencing (RNA-seq). Comparative transcriptome analysis revealed that knockdown of DDX17 led to extensive changes in gene expression profiles and ASEs in A549 cells that were confirmed in The Cancer Genome Atlas (TCGA)-LUAD dataset. These findings support the hypothesis that DDX17 may play a role in LUAD by

regulating the expression and AS of genes associated with apoptosis and cell proliferation. Therefore, a further understanding of the role and mechanism of action of DDX17 may provide new therapeutic strategies for LUAD patients.

## MATERIAL AND METHODS

All small interfering RNA (siRNA) duplexes were purchased from Gemma (Suzhou, China). The siRNA sequences were as follows: Nontargeting control siRNA (siNegative), 5′-UUCUCCGAACGUGUCACGUTT-3′ (sense); siRNA targeting DDX17 (siDDX17-3), 5′- GCACCCAUCCUUAUUGCUATT -3′ (sense).

### Cell culture and transfection

Human A549 lung adenocarcinoma cells were obtained from the China Center for Type Culture Collection (CCTCC; Wuhan, China). The identify of cell line was authenticated (STR, Isoenzymatic Assay, PCR). We cultured the A549 cells with 10% fetal bovine serum (FBS) in minimal essential medium (MEM). The cells were maintained at 37 °C under 5% $CO_2$ and were treated with 100 μg/ml streptomycin and 100 U/ml penicillin. Then the siRNA was transfected into the A549 cells with Lipofectamine 2000 (Invitrogen, Carlsbad, CA, USA) per the manufacturer's instructions. After 48 h of transfection, cells were harvested for reverse-transcription quantitative PCR (RT–qPCR) analysis.

### Assessment of gene expression

Glyceraldehyde-3-phosphate dehydrogenase (GAPDH) was used as an internal control. We synthesized the cDNA using standard methods, and we performed the RT-qPCR analysis on the Bio-Rad S1000 using the Bestar SYBR Green RT-PCR Master Mix (DBI Bioscience, Shanghai, China). Detailed primer sequences are listed in Table 1. Using GAPDH as a normalization factor, transcript expression was normalized and calculated by the 2-ΔΔCT method (*Livak & Schmittgen, 2001*). Statistical comparisons were made using paired Student's *t* test in GraphPad Prism 7.0 software (Graphpad Prism software; GraphPad, San Diego, CA, USA).

### Cell proliferation assay

Cell proliferation was examined using a Cell Counting Kit-8 (CCK-8; Dojindo, Co., Shanghai, China). In brief, A549 cells were seeded at a density of 6000 cells/well in 96-well culture plates. Cell-free wells were used as blank controls (blank), and cells treated with an equal volume of phosphate-buffered saline (PBS) were used as controls (control). After 24, 48 and 72 h of siRNA transfection, CCK-8 solution (20 μl) was added to the culture medium and incubated for an additional 3 h. Finally, the OD450 values were measured with a PerkinElmer EnVision plate reader. The cell proliferation rate was calculated according to the following formula: proliferation rate = (experimental OD value − blank OD value)/(control OD value − blank OD value) × 100%.

### Annexin V apoptosis assay

An Annexin V-fluorescein isothiocyanate (FITC) and propidium iodide (PI) detection kit (Shanghai Yeasen Corporation, Shanghai, China) was used to detect apoptosis according

**Table 1  Primers sets; related to the experimental procedures.**

| Gene | Primer | Sequence (5′–3′) | Related figures |
|---|---|---|---|
| GAPDH | Forward | GGTCGGAGTCAACGGATTTG | Fig. 2A |
| | Reverse | GGAAGATGGTGATGGGATTTC | |
| DDX17 | Forward | GGGTACCGCCTATACCTTCT | Fig. 2A |
| | Reverse | TGTTGGCTGAAGAAGTGGTC | |

**Notes.**
RT–qPCR primers used for gene expression verification.

to the manufacturer's instructions. Specifically, A549 cells were cultured in six-well plates for 24 h and were then transfected with siRNA for 24, 48, or 72 h. Cells were treated with an equal volume of PBS as a control. The treated and control cells were mixed gently, and 5 μl of Annexin V-FITC reagent and 10 μl of 20 μg/ml PI reagent were then added and incubated at room temperature for 10–15 min in the dark. Following incubation, 400 μl of PBS was added. Flow cytometric analysis was used to detect apoptosis within 1 h. Flow cytometric analysis was used to detect apoptosis within 1 h. Cells positive for Annexin V-FITC or negative for PI fluorescence were considered apoptotic cells.

## Acquisition of RNA-seq data from TCGA

For RNA-seq, LUAD gene-level expression data (526 tumor samples and 59 normal samples) from TCGA were downloaded from the UCSC Xena hub (http://xena.ucsc.edu). Splice junction data in BED format for 518 LUAD tumor and 59 normal samples from TCGA were downloaded from the Genomic Data Commons (GDC) portal for identification of ASEs (*Kahles et al., 2018*).

## RNA extraction and sequencing

A Total RNA Extractor TRIzol (Ambion, Life Technologies, Carlsbad, CA, USA) was used to extract total RNA. RQ1 DNase (Promega, Madison, WI, USA) was used to remove any remaining DNA from the total RNA. RNA quality and quantity were assessed by measuring the absorbance ratio at 260 nm/280 nm (A260/A280) using a Nanodrop One spectrophotometer (Thermo Scientific, San Jose, CA, USA). 1.5% Agarose gel electrophoresis was performed to ensure RNA integrity.

For each sample, 1 μg of total RNA was used as the input material for RNA-Seq library preparation using a Kapa Stranded mRNA-seq kit (KAPA Biosystems, Wilmington, MA, USA) according to manufacturer's instructions. We purified the polyadenylated mRNAs using VAHTS mRNA capture Beads (N401-01) and digested in fragmentation buffer. The fragmented mRNA was converted into double-stranded cDNA which was then subjected to end repair, poly(A) tailing and ligation with a KAPA Unique Dual-Indexed Adapter Kit (KAPA Biosystems, Wilmington, MA, USA). After ligation, the cDNA was sheared to fragments (300–500 bp). Finally, the DNA fragments were amplified, purified, quantified, and stored at −80 °C until used for sequencing. As the second strand of cDNA is synthesized, dTTP is replaced with dUTP, which quenches the second strand during amplification. The

libraries were prepared and were run on the Illumina Novaseq 6000 system to generate 150-bp paired-end reads following the manufacturer's instructions.

## Cleaning and alignment of raw RNA-seq data

To start, we discarded any raw read containing more than N bases. Then we used the ASTX-Toolkit (Version 0.0.13) to removed adaptors from raw sequencing reads and trimmed low-quality bases. Short reads containing fewer than 16 nt were discarded. Subsequently, clean reads were mapped to the GRCh38 genome by TopHat2, allowing no more than four mismatches (*Kim et al., 2013*). Based on uniquely mapped reads, read numbers and fragments per kilobase of transcript per million fragments mapped (FPKM) values were calculated for each gene (*Trapnell et al., 2010*).

## Analysis of differentially expressed genes (DEGs)

The analysis of DEGs was performed with the R Bioconductor package edgeR (*Robinson, McCarthy & Smyth, 2010*). The cutoff criteria for identifying DEGs were set as a *p* value <0.01 and a fold change>1.5 or <2/3.

To increase the statistical power, experiments were performed in three biological replicates. The statistical power of this experimental design, calculated in RNASeqPower (https://doi.org/doi:10.18129/B9.bioc.RNASeqPower) is 1 (Table S1).

## Analysis of AS

The ABLas pipeline was applied to define and quantify the ASEs and regulated ASEs (RASEs) between the samples according to previously described methods (*Jin et al., 2017a*; *Xia et al., 2017*). In brief, ten types of ASEs were identified by the ABLas pipeline on the basis of the splice junction reads: exon skipping (ES), alternative 5′ splice site selection (A5SS), alternative 3′ splice site selection (A3SS), intron retention (IR), mutually exclusive exons (MXE), mutually exclusive 5′UTRs (5pMXE), mutually exclusive 3′UTRs (3pMXE), cassette exon (CassetteExon), A3SS&ES and A5SS&ES.

For analysis of RBP-regulated ASEs, Student's *t* test was applied to evaluate the significance of the alteration ratio of ASEs. A false discovery rate (FDR) of 0.05 was set as the significance cutoff.

## Validation of DEGs and ASEs by RT–qPCR

To confirm the results of RNA-seq-based identification of DEGs in A549 cells, some DEGs were selected for verification by RT–qPCR. Information on the primers is shown in File S3. A549 cells transfected with DDX17 siRNA were harvested after 48 h for RT–qPCR analysis. cDNA was synthesized from RNA using M-MLV Reverse Transcriptase (Vazyme). RT–qPCR was carried out on a StepOne RealTime PCR System using HieffTM qPCR SYBR® Green Master Mix (Low Rox Plus; Shanghai Yeasen Corporation, Shanghai, China). The thermal cycling conditions used for PCR were as follows: denaturation at 95 °C for 5 min, followed by 40 cycles of denaturation at 95 °C for 15 s and annealing and extension at 60 °C for 30 s. PCR amplification was performed in triplicate for each sample. The RNA expression levels of all genes were normalized to the expression level of GAPDH.

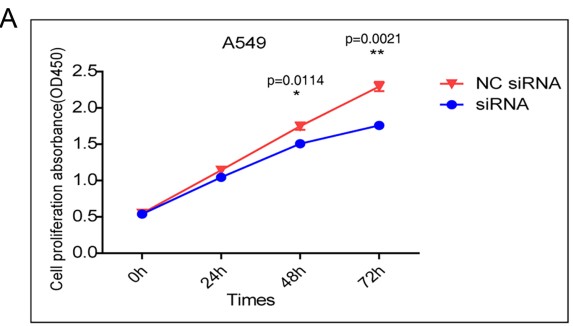
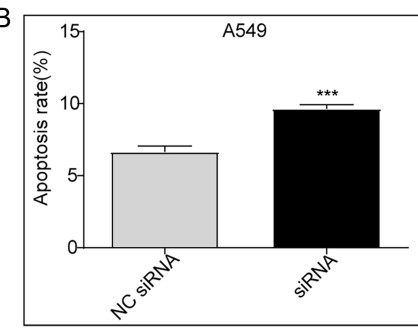

**Figure 1** **(A) A549 cells were transfected with the control vector or DDX17 knockdown plasmid.** A CCK-8 assay was performed to examine cell viability at 24, 48, and 72 h. (B) Effect of DDX17 knockdown on apoptosis regulation in A549 cells. Apoptosis was detected by an Annexin V-FITC/PI flow cytometric apoptosis assay. ***$p < 0.001$.

## Functional enrichment analysis

The biological functions of DEGs were comprehensively detected by Gene Ontology (GO) term and Kyoto Encyclopedia of Genes and Genomes (KEGG) pathway analysis using KOBAS 2.0 server (*Xie et al., 2011*). Hypergeometric test was used to test enrichment for each term and false discovery rate (FDR) procedure was used to correct *p* values.

## RESULTS

### Knockdown of DDX17 inhibits the proliferation and promotes the apoptosis of tumor cells

A CCK-8 assay was performed to evaluate cell proliferation. Compared to transfection with the control vector, siRNA knockdown of DDX17 significantly inhibited the proliferation of A549 cells (Fig. 1A). In addition, apoptosis analysis showed that DDX17 knockdown led to a marked increase in the apoptosis rate in A549 cells (Fig. 1B).

### DDX17 regulates the transcriptome in A549 cells

To examine DDX17-mediated transcriptional regulation in A549 cells, we constructed six cDNA libraries with three biological replicates each from DDX17 knockdown and control cells, which were subjected to RNA-seq analysis. DDX17 was knocked down by siRNA and quantified by RT–qPCR.

As shown in Fig. 2A, the expression of DDX17 was significantly reduced in A549 cells transfected with the targeting siRNA. Effective DDX17 knockdown was further confirmed by RNA-seq analysis and western blotting (Figs. 2B, 2C; Fig. S1). Principal component analysis (PCA) showed that the DDX17 siRNA group was obviously separated from the control group (Fig. 2D).

The DEGs, namely, 992 upregulated genes and 625 downregulated genes, are visualized in an MA plot (Fig. 2E). Heatmap analysis based on the expression patterns of the DEGs demonstrated clear separation between the DDX17 siRNA samples and control samples and high consistency among the three biological replicates (Fig. 2F). These results indicated that DDX17 markedly regulated the expression levels of many genes.

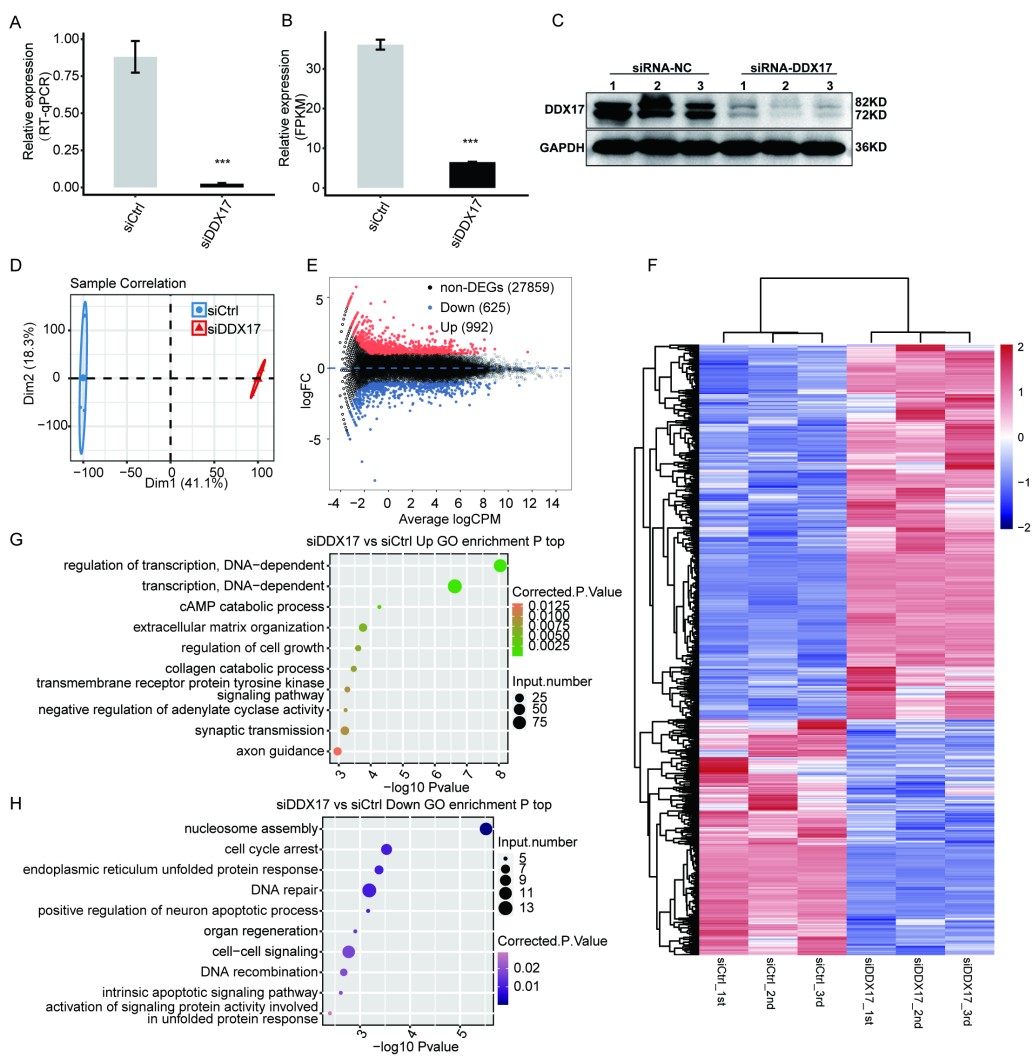

**Figure 2  DDX17 regulates the transcriptome in A549 cells.** DDX17 expression was quantified by RT–qPCR (A), RNA-seq (B), and Western blot (C) analyses. Original blots are presented in Fig. S1. ***$p <$ 0.001. (D) PCA of the two groups of samples based on normalized gene expression levels. (E) Identification of DDX17-regulated genes. In the MA plot, upregulated genes are labeled in red, whereas downregulated genes are labeled in blue. (F) Hierarchical clustering of DEGs in control and DDX17 knockdown samples. The FPKM values for each gene were log2-transformed and then median-centered. (G–H) Top 10 representative GO biological processes enriched with up- or downregulated genes.

We performed GO analyses to further investigate the potential biological functions of these DEGs (Figs. 2G, 2H). Functional enrichment analyses indicated that genes upregulated in the DDX17 knockdown group were enriched mainly in biological processes, including regulation of transcription (DNA-dependent), transcription (DNA-dependent), cAMP catabolic process, extracellular matrix organization, and regulation of cell growth, whereas downregulated genes were related primarily to nucleosome assembly, cell cycle arrest, DNA repair, endoplasmic reticulum unfolded protein process, and cell–cell signaling.

## DDX17 selectively regulates the expression of genes involved in the progression of cancer cells

C−X−C motif chemokine ligand 5 (CXCL5), C-C motif chemokine ligand 20 (CCL20), Growth differentiation factor 15 (GDF15), Stanniocalcin-2 (STC2) and RAD51 recombinase (RAD51) have been reported to play a role in promoting the proliferation, migration, and invasion of NSCLC cells (*Wang et al., 2018b*; *Wang et al., 2016*; *Wang et al., 2018a*; *Na et al., 2015*; *Hu et al., 2019*). As shown in Fig. 3 and Fig. S2, RNA-seq indicated that the mRNA expression of these genes was significantly reduced in the DDX17 knockdown group compared with the control group. To confirm the reliability of the RNA-seq data, RT−qPCR was performed. The qPCR and RNA-seq results exhibited a high degree of concordance.

Similarly, the expression of tumor suppressor genes (*e.g.*, Ankyrin repeat domain 1 (ANKRD1); hexamethylene-bis-acetamide-inducible protein 1 (HEXIM1); insulin like growth factor binding protein 4 (IGFBP4); insulin like growth factor binding protein 3 (IGFBP3); ras homolog gene family, member B (RHOB); and P53 apoptosis effector related to PMP22 (PERP)) was significantly increased in the DDX17 knockdown group compared with the control group (*Jiménez et al., 2017*; *Tan et al., 2016*; *Li et al., 2017b*; *Wang et al., 2017*; *Chen et al., 2016*; *Chen et al., 2011*). These results suggested that DDX17 may regulate the apoptosis and proliferation of LUAD cells by altering the expression of these genes.

## DDX17 selectively regulates AS of genes involved in cell proliferation, migration, invasion, and apoptosis

To investigate the impact of DDX17 knockdown on AS of genes in A549 cells, we evaluated splicing alterations in the RNA-seq data using the ABLas pipeline. A total of 2094 RASEs were detected (Fig. 4A). Among the nine types of ASEs, ES, A5SS, A3SS and CassetteExon were the main types detected. Hierarchical clustering of RASEs revealed obvious separation between the DDX17 siRNA samples and the control samples, with a high degree of consistency among the three biological replicates (Fig. 4B). To explore the possible association between the regulated alternatively spliced genes (RASGs) and DEGs, we examined the overlap between the two datasets (Fig. 4C). Both the expression level and AS of 71 genes were significantly affected by DDX17 knockdown.

GO functional enrichment analysis showed that RASGs were enriched mainly in biological processes related to the terms apoptotic process, DNA repair, cellular protein metabolic process, viral reproduction, positive regulation of translation, and positive regulation of apoptotic process (Fig. 4D). Among these RASGs, the following have previously been reported to mediate the proliferation, migration, invasion, and apoptosis of cancer cells: Ribosomal protein S27a (RPS27A); LSM1 homolog, mRNA degradation associated (LSM1); legumain (LGMN); inverted formin 2 (INF2); LDL receptor related protein 5 (LRP5); and filamin A (FLNA) (*Yu et al., 2021*; *Watson et al., 2008*; *Roslan et al., 2019*; *Zhang & Lin, 2021*; *Qian et al., 2018b*; *Qian et al., 2018a*). Significantly fewer ES events in LSM1, CassetteExon events in INF2, CassetteExon events in LGMN and A3SS events in LRP5 were detected by RNA-seq in the DDX17 knockdown group than in the control group (Figs. 4E, 4F; Fig. S3). Conversely, significantly more A5SS events in RPS27A

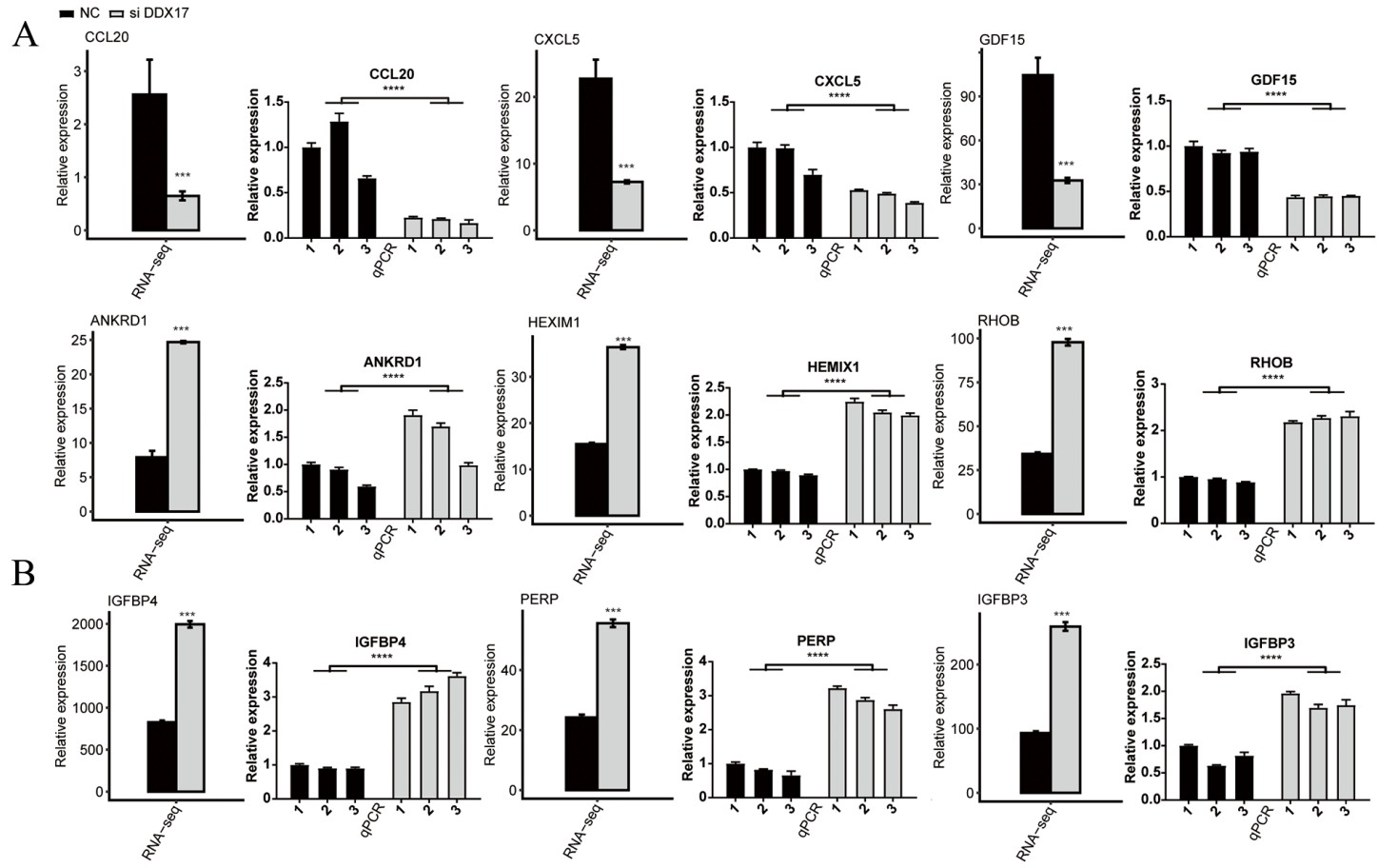

**Figure 3** The mRNA expression levels of DDX17-regulated genes related to (A) cell proliferation, migration, invasion and (B) apoptosis from the RNA-seq data and RT–qPCR validation. The error bars indicate the means ± SEMs. ***$p < 0.001$. ****$p < 0.0001$.

and CassetteExon events in FLNA were detected in the DDX17 knockdown group than in the control group. These results were confirmed by RT–PCR. In particular, these findings indicate that DDX17 can regulate both the AS and mRNA expression of LGMN.

## DDX17 regulates the transcriptome in LUAD in the TCGA database

To further confirm the transcriptional activity of DDX17, we analyzed gene expression data from TCGA datasets (including 526 LUAD samples and 59 normal lung samples) by using the UCSC Xena platform (https://xena.ucsc.edu/). We selected 20 samples with the highest DDX17 expression (all of which are cancer tissues) and 20 samples with the lowest DDX17 expression (including 19 cancer tissues and 1 normal lung tissue) to assess the transcriptional regulation of DDX17. A total of 5,243 genes were differentially regulated, among which 4,218 were upregulated and 1,025 genes were downregulated (Fig. 5A). The DEGs were visualized in a heatmap, which showed clear separation between the two groups (Fig. 5B). The Venn diagram of the DEGs showed that 67 genes were upregulated in LUAD samples with high DDX17 expression compared to those with low DDX17 expression, whereas these genes were downregulated in A549 cells after DDX17 knockdown (Fig. 5C).
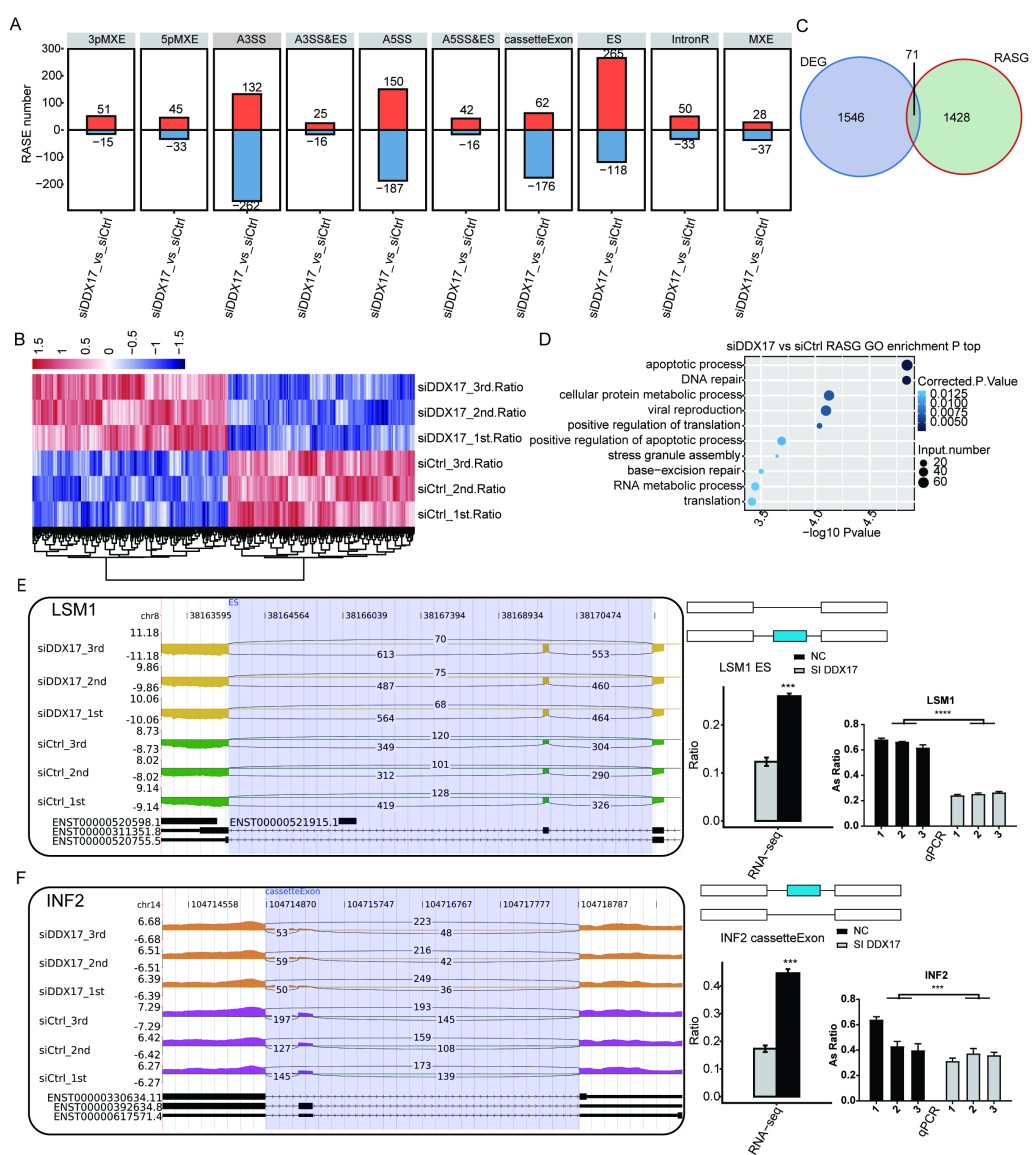

**Figure 4  DDX17 regulates AS in A549 cells.** (A) Frequency distribution of different types of DDX17-regulated ASEs. (B) Hierarchical clustering of RASGs in control and DDX17 knockdown samples. The FPKM values for each gene were log2-transformed and then median-centered. (C) Venn diagram showing the numbers of overlapping RASGs and DEGs. (D) Top 10 GO biological processes enriched with RASGs. (E–F) DDX17 regulates AS of LSM1 and INF2. The Integrated Genomics Viewer (IGV) sashimi plots show AS changes in DDX17 knockdown cells and control cells (left half of each plot), and the transcripts of the genes are shown below the sashimi plots. The schematic diagrams show the structures of the ASEs (upper-right half of each plot). The constitutive exon sequences are denoted by white boxes, the intron sequences by horizontal lines, and the alternative exons by blue boxes. RNA-seq quantification and RT–qPCR validation of ASEs are shown at lower-right half of each plot. The alteration ratio of ASEs in the RNA-seq data was calculated using the following formula: alternative splice junction reads/(alternative splice junction reads + model splice junction reads). Student's t test was performed to compare DDX17 knockdown and control cells, with a p value of less than 0.05 considered to indicate significance. ***$p < 0.001$. ****$p < 0.0001$.

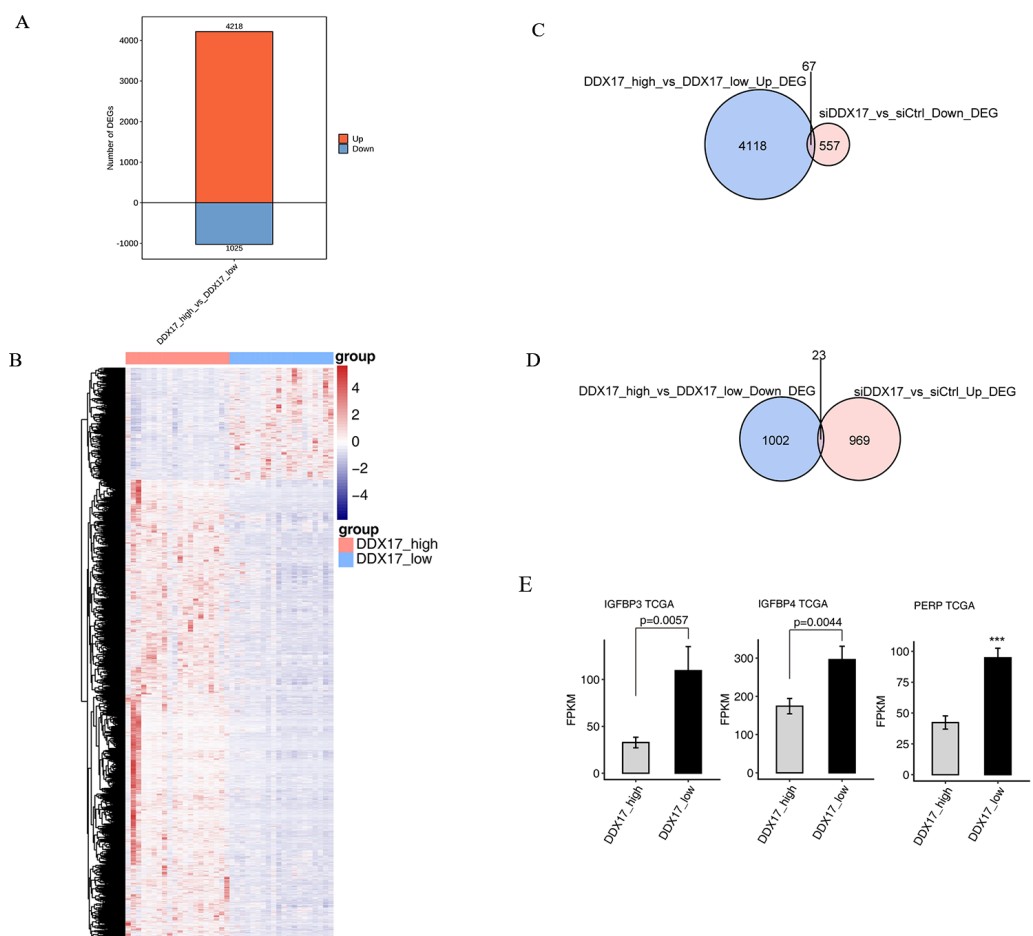

**Figure 5 The DDX17 regulated transcriptome in LUAD of TCGA database.** (A) Identification of DDX17 regulated genes. Up-regulated genes are labeled in red, whereas down-regulated are labeled in blue in the bar plot. (B) Hierarchical clustering of DEGs in high and low expression of DDX17 samples. FPKM values are log2-transformed and then median-centered by each gene. (C–D) Venn diagram showing the overlap genes number of up or down DEGs in LUAD and RNA-seq. (E) Genes expression of DDX17 regulated cell apoptosis genes from LUAD of TCGA database. ***p < 0.001.

Similarly, 23 genes were downregulated in LUAD samples with high DDX17 expression compared to those with low DDX17 expression, whereas these genes were upregulated in A549 cells after DDX17 knockdown (Fig. 5D). Among these 23 genes, apoptosis-related genes such as IGFBP4, IGFBP3 and PERP were upregulated in LUAD samples with low DDX17 expression compared to those with high DDX17 expression in the TCGA database (Fig. 5E).

## DDX17 regulates AS of genes in LUAD in the TCGA database

Finally, to investigate whether DDX17 influences AS of genes in LUAD patients, we analyzed BED format splicing data (including 518 LUAD samples and 59 normal lung samples) from the GDC portal (*Kahles et al., 2018*). We selected 20 samples with the highest DDX17 expression and 20 samples with the lowest DDX17 expression to assess ASEs regulated

He et al. (2022), *PeerJ*, DOI 10.7717/peerj.13895

11/21

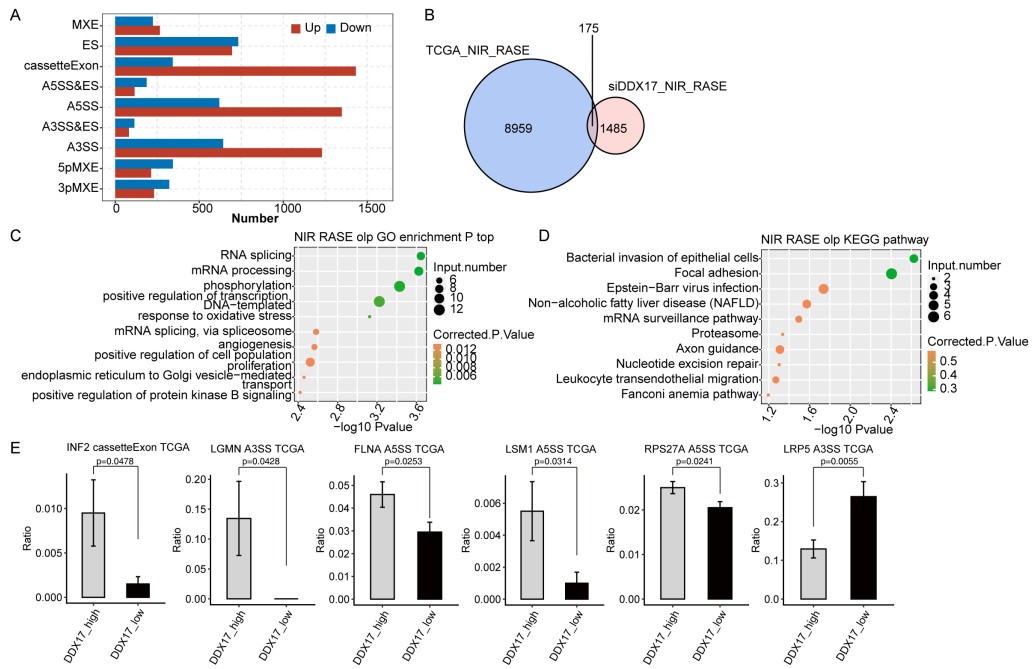

**Figure 6  DDX17 regulates AS of genes in LUAD in the TCGA database.** (A) Frequency distribution of different types of DDX17-regulated ASEs in samples with high and low DDX17 expression. (B) Venn diagram showing the numbers of overlapping RASGs in TCGA-LUAD data and RNA-seq data. (C) Top 10 GO biological processes enriched with the overlapping RASGs in TCGA-LUAD data and RNA-seq data. (D) The top 10 KEGG pathways of overlapping RASGs in TCGA-LUAD data and RNA-seq data. (E) DDX17 regulates AS of genes. The alteration ratio of ASEs in the RNA-seq data was calculated using the following formula: alternative splice junction reads/(alternative splice junction reads + model splice junction reads). Student's *t* test was performed to compare LUAD samples with high and low expression of DDX17, with a p value of less than 0.05 considered to indicate significance.

by DDX17. Similar to the observations in A549 cells, A5SS, A3SS and CassetteExon were the main types of ASEs regulated by DDX17 in LUAD patients (Fig. 6A). By analyzing the overlap between the RASEs in the TCGA database and the RASEs in siDDX17 cells, we found significant overlap (175 genes) between the RASGs in the LUAD data and the RNA-seq data (Fig. 6B). GO functional enrichment analysis indicated that these genes were enriched mainly in biological processes related to the terms RNA splicing; mRNA processing; phosphorylation; positive regulation of transcription (DNA-templated); mRNA splicing, via spliceosome; and proliferation (Fig. 6C). Then, these overlapping genes were subjected to KEGG pathway enrichment analysis and were found to be enriched mainly in biological processes related to bacterial invasion of epithelial cells, focal adhesion, Epstein−Barr virus infection, the mRNA surveillance pathway, and the proteasome (Fig. 6D). Among the overlapping genes, significantly fewer A5SS events in LSM1, CassetteExon events in INF2, A5SS events in FLNA, A5SS events in RPS27A, and A3SS events in LGMN were detected by RNA-seq in LUAD samples with low DDX17 expression than in those with high DDX17 expression, whereas A3SS events in LRP5 were more prevalent in LUAD samples with high DDX17 expression (Fig. 6E).

## DISCUSSION

The DEAD-box RNA helicases are the largest family of RNA helicases that play a significant role in all aspects of RNA metabolism, including the regulation of gene expression and AS (*Sarkar & Ghosh, 2016*; *Linder & Jankowsky, 2011*; *Zhang & Li, 2021*). DDX17, a DEAD-box ATPase, is highly expressed in many types of human cancers, where it participates in tumor initiation, progression, and metastasis (*Wu, 2020*; *Xue et al., 2019*; *Shin et al., 2007*; *Luo et al., 2020*). However, the role of DDX17 in LUAD is incompletely determined.

In this study, we showed that siRNA knockdown of DDX17 significantly inhibited proliferation and led to increased apoptosis in A549 cells. This oncogenic effect of DDX17 in LUAD is consistent with the results of studies on other cancer types (*Xue et al., 2019*; *Shin et al., 2007*; *Wu et al., 2021*).

We next examined DDX17-mediated transcriptional regulation in A549 cells and identified 1617 DEGs, namely, 992 upregulated DEGs and 625 downregulated DEGs. GO functional enrichment analysis indicated that the downregulated genes, such as CXCL5, CCL20 and RAD51, in the DDX17 knockdown group were enriched mainly in nucleosome assembly, cell cycle arrest, DNA repair, endoplasmic reticulum unfolded protein process, and cell–cell signaling.

CXCL5, a member of the CXC-type chemokine family, is overexpressed in many cancers and can promote tumor cell proliferation, migration and invasion (*Li et al., 2011*; *Zhou et al., 2014*; *Mao et al., 2020*; *Zhao et al., 2017*). In addition, high expression of CXCL5 was found to be associated with poor tumor differentiation and poor survival of NSCLC patients (*Wang et al., 2018b*). CCL20 can promote lung cancer cell proliferation and migration by activating the ERK and PI3K signaling pathways (*Wang et al., 2016*). RAD51 is an ATPase that is essential for the repair of DNA damage (*Bonilla et al., 2020*). High expression of RAD51 is an indicator of adverse prognosis in NSCLC patients (*Qiao et al., 2005*). In KRAS-mutant lung cancer cells, RAD51 may facilitate DNA damage repair and promote cell survival (*Hu et al., 2019*).

In contrast, the upregulated genes, such as IGFBP4, IGFBP3, and PERP, in the DDX17 knockdown group were enriched mainly in biological processes, including regulation of transcription (DNA-dependent), transcription (DNA-dependent), and regulation of cell growth. To further confirm transcriptional regulation by DDX17 in clinical samples, we compared 20 samples with the highest DDX17 expression with 20 samples with the lowest DDX17 expression in the TCGA-LUAD dataset. The expression of some overlapping genes was affected by DDX17 interference both in LUAD samples and in A549 cells. Among these overlapping genes, IGFBP4, IGFBP3 and PERP were significantly upregulated both in LUAD samples with low DDX17 expression and in DDX17 knockdown A549 cells.

PERP is a plasma membrane protein and a key mediator of p53-mediated apoptotic pathways (*Ihrie et al., 2003*). By inducing apoptosis and suppressing VEGF expression, PERP gene therapy was found to attenuate LUAD cell growth in a human lung cancer xenograft model (*Chen et al., 2011*). IGFBP3 and IGFBP4 are members of the insulin-like growth factor (IGF) binding protein family, whose members can bind to IGFs and regulate cell proliferation and apoptosis (*Brahmkhatri, Prasanna & Atreya, 2015*). IGFBP4

overexpression can inhibit tumor proliferation and induce apoptosis by increasing the expression level of the proapoptotic protein BAX and decreasing that of the antiapoptotic protein BCL2 in A549 lung cancer cells (*Wei et al., 2021*). IFGBP3 overexpression may induce lung cancer cell apoptosis and increase cisplatin sensitivity by suppressing IGF1 signaling (*Wang et al., 2017*). Further studies are warranted to explore whether DDX17 can regulate the survival of lung cancer cells by regulating the expression of these genes.

DDX17 is an important regulator of AS, and its deregulation is considered a hallmark of cancer and a potential therapeutic target (*Hönig et al., 2002*; *Urbanski, Leclair & Anczuków, 2018*). DDX17 and DDX5 may play a critical role in tumor cell migration by simultaneously regulating the transcriptional activity and AS of NFAT5 in HeLa cells (*Germann et al., 2012*). In addition, DDX17 regulates AS of PXN-AS1, leading to the production of a novel transcript that promotes HCC metastasis (*Zhou et al., 2021*).

In the present study, numerous DDX17-regulated ASEs were observed both in A549 cells and in LUAD patients in the TCGA dataset. GO functional enrichment analysis showed that the RASGs were enriched mainly in biological process terms related to apoptotic process, DNA repair, cellular protein metabolic process, viral reproduction, positive regulation of translation, and positive regulation of apoptotic process. These ASEs in LSM1, INF2, LGMN, LRP5, RPS27A, and FLNA were significantly affected by DDX17 in A549 cells, and this finding was confirmed in LUAD samples in TCGA datasets.

LGMN is highly expressed in various tumors and can promote tumor progression and invasion (*Zhen et al., 2015*). Inhibition of LGMN may be a potent strategy for cancer therapy (*Zhao et al., 2021*). Our findings showed that DDX17 knockdown significantly affected both the gene expression and AS of LGMN in A549 cells. However, the type of ASE (CassetteExon) in LGMN affected by DDX17 knockdown in A549 cells was inconsistent with the type (A3SS) identified in the TCGA-LUAD dataset.

INF2 is a member of the formin family of proteins, which regulate the polymerization of actin (*Zhao et al., 2022*). INF2 may induce apoptosis or act as an oncogene in different cancers (*Qian et al., 2018b*; *Heuser et al., 2018*; *Jin et al., 2017b*; *Heuser et al., 2020*). In the present study, knockdown of DDX17 significantly reduced CassetteExon events in INF2 in A549 cells, and this result was confirmed in the TCGA-LUAD dataset. Therefore, we hypothesized that DDX17 may promote LUAD cell proliferation by regulating AS of INF2. This hypothesis is currently being tested in our laboratory.

## CONCLUSIONS

DDX17 can regulate both AS and transcription in LUAD cells. DDX17 knockdown significantly inhibits proliferation and induces apoptosis in A549 cells. Furthermore, we showed that DDX17 can regulate the expression or AS of some cancer-related genes and speculated that INF2 may be a target gene of DDX17 in regulating AS in LUAD. Further studies are required to investigate whether DDX17-mediated signaling contributes to the tumorigenesis of LUAD.

### Funding

This study was supported by the Beijing Science and Technology Innovation Medical Development Foundation (No. KC-2021-JX-0186-30). The funders had no role in study design, data collection and analysis, decision to publish, or preparation of the manuscript.

### Grant Disclosures

The following grant information was disclosed by the authors:

Beijing Science and Technology Innovation Medical Development Foundation: KC-2021-JX-0186-30.

### Competing Interests

The authors declare there are no competing interests.

### Author Contributions

- Cheng He conceived and designed the experiments, analyzed the data, authored or reviewed drafts of the article, and approved the final draft.
- Gan Zhang performed the experiments, analyzed the data, prepared figures and/or tables, and approved the final draft.
- Yanhong Lu performed the experiments, prepared figures and/or tables, and approved the final draft.
- Jingyue Zhou performed the experiments, analyzed the data, authored or reviewed drafts of the article, and approved the final draft.
- Zixue Ren conceived and designed the experiments, authored or reviewed drafts of the article, and approved the final draft.

### Data Availability

Data is available at NCBI GEO: GSE196835.

### Supplemental Information

Supplemental information for this article can be found online at http://dx.doi.org/10.7717/peerj.13895#supplemental-information.

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
