# Peer review of "DDX17 modulates the expression and alternative splicing of genes involved in apoptosis and proliferation in lung adenocarcinoma cells"

_PeerJ, doi:10.7717/peerj.13895_

## Round 0.1 · original submission · Major Revisions

Your manuscript was considered interesting and valuable by the reviewers since it investigates the role of DDX17 in cancer, which is not very well characterized. However, the reviewers identified a number of issues that need to be addressed. They suggested that tables of the DEGs you identified be provided and that your raw RNA-seq data be deposited into a repository. Furthermore, the reviewers requested additional detail regarding some figures, specifically to indicate which genes promote proliferation, and which promote apoptosis in figure 3 and what criteria were used to select these genes. Furthermore, the reviewers pointed out that some of the expression levels of the genes shown in figure 3 show changes in expression smaller than two-fold (STC2 and RAD51), so they do not meet the criteria for differential regulation. Additionally, the reviewers noted that the p-values shown in figures 1C and 5F are >0.05, so they do not meet the threshold for statistical significance. Finally, the reviewers would like you to explain how you reached the conclusion that INF2 is a target of DDX17, as well as the rationale underlying your conclusion that DDX17 regulates alternative splicing of the genes shown in figure 6, rather than increased DDX17 expression being associated with alternative splicing of these genes.

Please, submit a detailed rebuttal which shows where and how you have taken all comments and suggestions into consideration. If you do not agree with some of the reviewers’ comments or suggestions, please explain why. Your rebuttal will be critical in making a final decision on your manuscript. Please, note also that your revised version may enter a new round of review by the same or by different reviewers. Therefore, I cannot guarantee that your manuscript will eventually be accepted.

·

Basic reporting

No comment

Experimental design

No comment

Validity of the findings

No comment

Additional comments

No comment

Reviewer 2 ·

Basic reporting

Nowadays, the roles of DEAD-box RNA-binding protein (RBP) DDX17 in cancer are not well-explained and need to be further explored. Accordingly, this research group explained the biological functions of DDX17 in the proliferation, migration,
invasion, and apoptosis of lung adenocarcinoma cells. This group demonstrated that
DDX17 knockdown increased proapoptotic genes and decreased the pro-proliferative genes. More importantly, this group found that aberrantly expressed DDX17 displayed significant correlations with several alternative splicing biomarkers in lung adenocarcinoma cells. After carefully revised, this report could be suite for this journal.

1. In Fig.1C, the association between the DDX17 expression level and OS in patients with LUAD in the TCGA database was evaluated using KM survival analysis. However, the p-values > 0.05. Please re-analyze or replace the data.
2. The DEGs, namely, 992 upregulated genes and 625 downregulated genes, are visualized in
239 an MA plot (Fig. 2E). Please provided the DEGs tables. More importantly, the raw-data of RNA-seq, Fig.2E-F, Fig. 4C-D, Fig. 5B-D, Fig. 6B-D and others should be uploaded as the supplementary tables.
3. In Fig.3, What does the x-axis (1,2,3, etc) mean? And Please note 4 stars in statistics in Figure Legends.
4. KM survival analysis showed that high IGFBP4 expression was associated with better OS in LUAD patients in the TCGA database. This result is Fig.5F? However, the p-values > 0.05. Please re-analyze or replace the data.
5. In Conclusions, why authors demonstrated “INF2 may be a target gene of DDX17”?

Experimental design

NONE

Validity of the findings

NONE

Additional comments

NONE

Annotated reviews are not available for download in order to protect the identity of reviewers who chose to remain anonymous.

Reviewer 3 ·

Basic reporting

The manuscript is clear and professional with sufficient background introduction and literature support.
The article structure and figure are logical but there are some minor issues with the figure 3 arrangement. This manuscript includes all results relevant to the hypothesis.

Experimental design

no comment

Validity of the findings

no comment

Additional comments

1. Apoptosis and proliferation marker genes should be identified. Fig.3 should be rearranged in a simple way.

2. Line 296-298, which samples did the author use (LUAD or normal samples or mixed), and how to compare them?

3. Discussion part, the mechanism of DDX17-induced apoptosis and proliferation is not clear. The author just listed the potential genes that may affect apoptosis or proliferation. The author should explain why to select these genes for further analysis since there is no evidence showing that apoptosis is involved in the processes (Fig. 5C-D). Was it based on gene enrichment or literature?

Reviewer 4 ·

Basic reporting

The paper is well written in professional English with sufficient details provided. The data is focused on the hypothesis and the results are well illustrated.

Some minor points need to be corrected. For example, in line 114, "glyceraldehyde 3-phosphate
115 dehydrogenase" needs to be replaced by "GAPDH", as the abbreviation was mentioned before.

Experimental design

The experiments are well explained. All the details are provided.

Validity of the findings

There are some problems that need to be corrected/clarified.
1. Figure 1C No legend is provided for the 4 dot lines.
2. In figure 3, the author chose some of the genes to evaluate in RT-PCR analysis. Are those genes the only genes that promote cell proliferation, migration, and invasion? If not, why these genes were picked for downstream analyses?

Two major problems:
1) in Figure 3: due to the nature of PCR-based analysis, any differences that are less than 2 folds are not accurate. Therefore, genes like RAD51 and STC2 are not considered as differentially expressed although they matched the mathematical criteria.

2) in Figure 6, the author demonstrated that DDX17 regulate splicing event as found in the TCGA dataset. Is any of these events can be found in the A549 cell lines with/without DDX17 siRNA? If not, the findings in the TCGA database only suggest an association between DDX17 expression and those events, but not a causal effect. Thus, the author can not make the conclusion that DDX17 modulates gene splicing.

---

## Round 0.2 · accepted · Accept

Thank you for thoroughly addressing the reviewers' comments and thus improving your manuscript.

Reviewer 2 ·

Basic reporting

Good

Experimental design

Good

Validity of the findings

Good

Additional comments

None.

Reviewer 3 ·

Basic reporting

The authors have addressed the reviewers' concerns.

Experimental design

no comment

Validity of the findings

no comment

Additional comments

no comment

Reviewer 4 ·

Basic reporting

The author addressed all my questions.

Experimental design

N/A

Validity of the findings

N/A

Additional comments

N/A